# Integrative proteomics in prostate cancer uncovers robustness against genomic and transcriptomic aberrations during disease progression

Leena Latonen[1,2], Ebrahim Afyounian [1], Antti Jylhä[3], Janika Nättinen[3], Ulla Aapola[3], Matti Annala [1], Kati K. Kivinummi[1], Teuvo T.L. Tammela[4], Roger W. Beuerman[3,5,6,7,8], Hannu Uusitalo[3,9], Matti Nykter[1,10] & Tapio Visakorpi[1,2]

To understand functional consequences of genetic and transcriptional aberrations in prostate cancer, the proteomic changes during disease formation and progression need to be revealed. Here we report high-throughput mass spectrometry on clinical tissue samples of benign prostatic hyperplasia (BPH), untreated primary prostate cancer (PC) and castration resistant prostate cancer (CRPC). Each sample group shows a distinct protein profile. By integrative analysis we show that, especially in CRPC, gene copy number, DNA methylation, and RNA expression levels do not reliably predict proteomic changes. Instead, we uncover previously unrecognized molecular and pathway events, for example, several miRNA target correlations present at protein but not at mRNA level. Notably, we identify two metabolic shifts in the citric acid cycle (TCA cycle) during prostate cancer development and progression. Our proteogenomic analysis uncovers robustness against genomic and transcriptomic aberrations during prostate cancer progression, and significantly extends understanding of prostate cancer disease mechanisms.

[1] Prostate Cancer Research Center, Faculty of Medicine and Life Sciences and BioMediTech Institute, University of Tampere, Tampere 33014, Finland. [2] FimLab Laboratories, Tampere University Hospital, Tampere 33101, Finland. [3] Department of Ophthalmology, Faculty of Medicine and Life Sciences, University of Tampere, Tampere 33014, Finland. [4] Department of Urology, University of Tampere and Tampere University Hospital, Tampere 33521, Finland. [5] Singapore Eye Research Institute, Singapore 169856, Singapore. [6] Duke-NUS Neuroscience, Singapore 169857, Singapore. [7] Duke-NUS Medical School Ophthalmology and Visual Sciences Academic Clinical Program, Singapore 169857, Singapore. [8] Ophthalmology, Yong Loo Lin Medical School, National University of Singapore, Singapore 119228, Singapore. [9] Tays Eye Centre, Tampere University Hospital, Tampere 33521, Finland. [10] Science Center, Tampere University Hospital, Tampere 33521, Finland. These authors contributed equally: Ebrahim Afyounian, Antti Jylhä, Janika Nättinen. Correspondence and requests for materials should be addressed to M.N. (email: matti.nykter@uta.fi) or to T.V. (email: tapio.visakorpi@uta.fi)

Prostate cancer is the most common male malignancy in Western countries, and the second most common cancer among men overall[1]. Currently, no curative treatment exists for castration resistant prostate cancer (CRPC)[2]. To understand the etiology of the disease and to find more specific drug targets, the driver mutations and expressional changes in prostate cancer have been examined through extensive genomic and transcriptomic characterization[3–7]. Although significant insight has been gained through these efforts, it is clear that not all molecular alterations influencing the tumor outcome can be captured through these approaches.

Proteins are regulated at multiple levels, and their expression is not always reflecting the levels of mRNA[8,9]. Thus, a comprehensive understanding of the molecular events in cancer require thorough investigation of the proteome[10]. Recent developments in mass spectrometric methods[11–13] have enabled high throughput analysis of clinical patient samples, and the first integrative studies involving large scale, mass spectrometry-based proteomics of human cancer have recently been published[14–16]. For prostate cancer, recent proteomic advancements have included high scale, mass spectrometry-based studies performed in diagnostic body fluids[17,18], as well as primary tumors[19] and the tumor microenvironment[20]. So far, the only integrative proteogenomic analysis of clinical prostate cancer involved genomic and transcriptomic data of CRPC combined with phosphoproteomic analysis[21]. Despite the merits of this study in interrogating the active signaling pathways in CRPC, the large-scale proteomic view of PC and CRPC, and reflections of them to the disease progression are still lacking.

Here, we provide the first integrative view on human prostate cancer with the proteome of clinical patient samples of benign prostatic hyperplasia (BPH), untreated primary prostate cancer (PC) and locally recurrent CRPC. Our analysis adds a new level to the current knowledge of prostate cancer development and

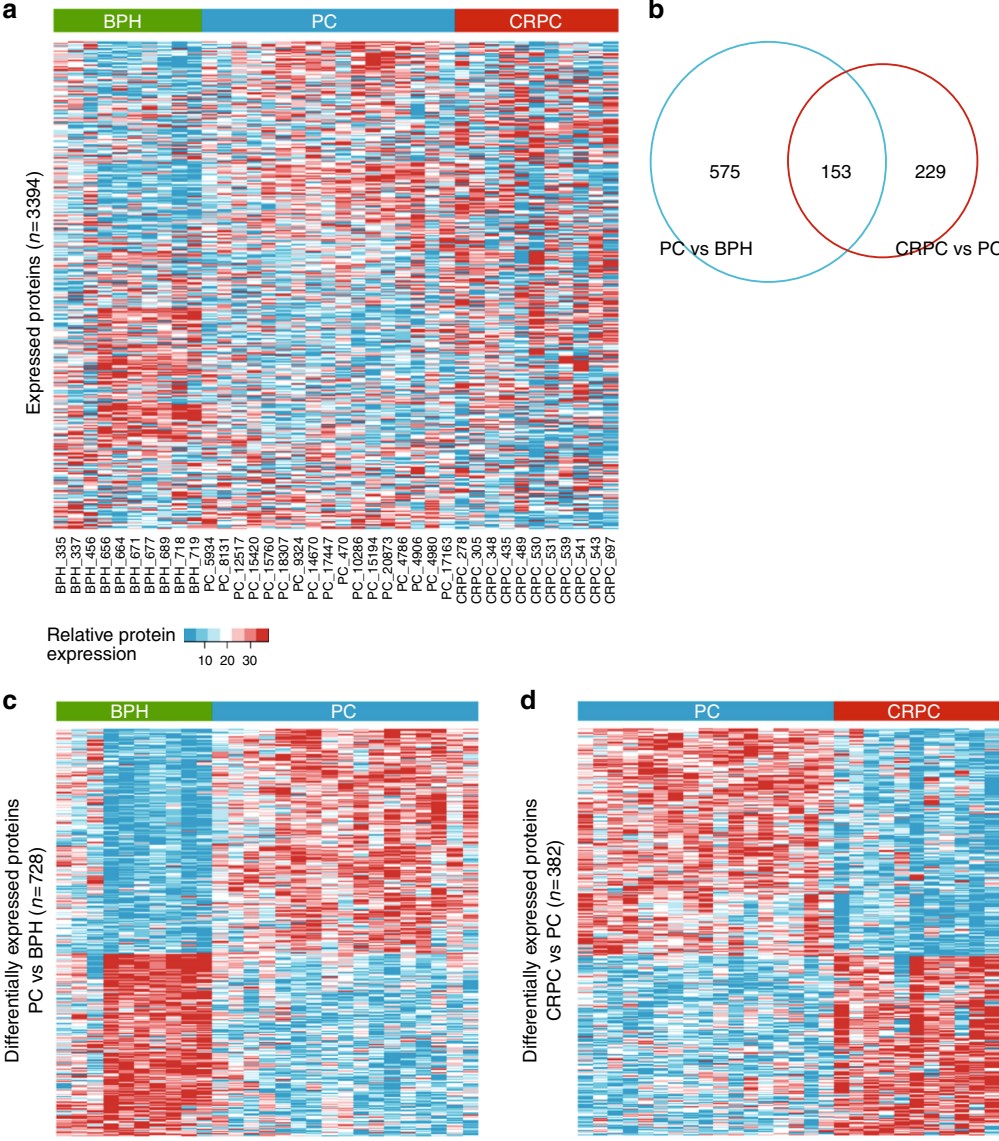

**Fig. 1** Proteomic analysis reveals distinct protein expression patterns in PC and CRPC. **a** Heat map of all protein expressions identified and quantified by mass spectrometry in the proteomic analysis of BPH and prostate cancer samples (PC and CRPC). Each column of heat map represents a patient sample and each row represents a specific protein (n = 3394). **b** Venn diagram showing the numbers of differentially expressed proteins in PC vs BPH and CRPC vs PC comparisons. Only a minority of the differentially expressed proteins overlap between the comparisons. **c, d** Heat maps of the differentially expressed proteins in **b** show clearly distinctive patterns of protein expression between disease groups. PC compared to BPH samples (n = 728) is shown in **c**, and CRPC compared to PC samples (n = 382) is shown in **d**. Color key of relative expression in **a** applies also to **c** and **d**

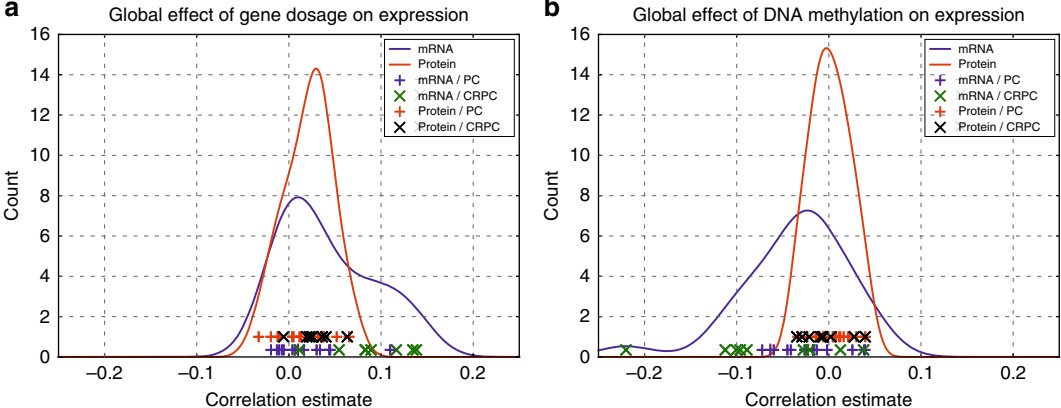

**Fig. 2** Global expression changes associated with gene copy number and DNA methylation are visible at the transcriptomic but not at proteomic level. **a** Correlation distributions of mRNA and protein expression with gene copy number. Lines represent effects in all analyzed genes in all samples, and show that gene dosage has higher positive correlation with mRNA expression than protein expression in prostate cancer on a global scale. Symbols on the bottom of the graph represent individual samples, and show how most of the CRPC samples have a higher positive correlation compared to PC samples at the mRNA level, as at the protein level no such difference between the disease groups is observed. **b** Correlation distributions of mRNA and protein expression with DNA methylation. Lines represent effects in all analyzed genes in all samples, and show that DNA methylation has higher negative correlation with mRNA expression than protein expression in prostate cancer on a global scale. Symbols on the bottom of the graph represent individual samples, and show how most of the CRPC samples have a decreased correlation compared to PC samples at the mRNA level, as at the protein level no such difference between the disease groups is observed

progression by identifying several molecular and pathway events not previously described based on transcriptomic data.

## Results

**Mass spectrometric analysis of proteomic profiles.** Samples of 10 BPH, 17 untreated PC (Supplementary Table 1), and 11 CRPC (Supplementary Table 2) were analyzed. The CRPC samples came from patients that had been treated either by castration and/or antiandrogens and experienced urethral obstruction (ie. local recurrence) during the treatment. With sequential window acquisition of all theoretical fragment ion spectra mass spectrometry (SWATH-MS), we identified a total of 213,979 peptides, corresponding to 1,753,161 identified spectra in an assembly of 4601 protein groups using false discovery rate of 1%. Protein and peptide quantification data can be found from Supplementary Data 1. From this library, 3394 proteins had distinct peptides sequences with matching spectras to SWATH-MS analysis and were quantified in all samples (Supplementary Data 2). The SWATH-MS data was reproducible with mean intraclass correlation (ICC) coefficient of 0.98 between technical replicate MS analyses. Permutation tests (Spearman correlation) showed that 98.6% of the technical replicate MS analyses had a $p$-value < 0.05, demonstrating excellent quality. The represented protein classes (PANTHER protein class) and gene ontology groups (GO; molecular functions, cellular components, and biological processes) are shown in Supplementary Fig. 1a. The distribution of the proteins into different protein classes was largely according to expected as compared to *Homo sapiens* reference list (Supplementary Fig. 1a,b). The major overrepresented groups included the highly abundant nucleic acid binding (mainly RNA binding) and ribosomal proteins, oxidoreductases, and hydrolases. The major underrepresented groups were transcription factors and receptors, including immunoglobulins, consistent with the cell type-dependent expression of especially the latter group.

Expression profiles of the identified proteins in the prostate tissue samples are shown in Fig. 1a. We wanted to assess changes occurring at the protein level during prostate cancer development and progression. As a model for benign tissue, we used BPH samples, against which primary PC samples were compared to

identify early cancerous events. To identify events related to cancer progression and castration resistance, CRPC samples were compared to PC samples. We identified 728 proteins in PC vs BPH and 382 proteins in CRPC vs PC to be differentially expressed (Wilcoxon rank sum test with Benjamini & Hochberg adjustment $p$-value < 0.05 and median ratio (fold change) >1.5) between the comparison groups (Fig. 1b). While the overall protein classes of the differentially expressed proteins and their distribution to groups of molecular function, cellular component, and biological process were similar between PC vs BPH and CRPC vs PC comparisons (assessed by Panther analysis; data not shown), only a subset ($n = 153$) of the differentially expressed proteins were common between the comparison groups (Fig. 1b). The expression profiles of the differentially expressed proteins clearly distinguished between the patient sample groups, as shown in Fig. 1c (PC compared to BPH) and Fig. 1d (CRPC compared to PC). These results show that the proteomic profile of prostate cancer is significantly altered during the course of the disease.

**Correlations of copy number and methylation with proteomics.** We have previously performed whole genome sequencing for copy number analysis, DNA methylation sequencing, and whole transcriptome sequencing to majority of the samples used in the proteomic analysis described here (Supplementary Table 3) [7,22]. We compared the correlation between gene copy number, and mRNA or protein expression levels between the common samples. While at the transcriptome level, the mRNA expression and copy number have an increased overall correlation in the CRPC samples compared to PC samples (Fig. 2a, Supplementary Fig. 2a), a similar global correlation change with gene copy number is not present at the proteomic level. Next, we compared the correlation between DNA methylation at differentially methylated regions (DMRs), and mRNA or protein expression levels in the same samples. Similarly as with the copy number data, the increased negative correlation between DNA methylation and mRNA expression at a global level in the CRPC samples compared to PC samples is not detected at the level of the proteome (Fig. 2b, Supplementary Fig. 2b). These results suggest that,

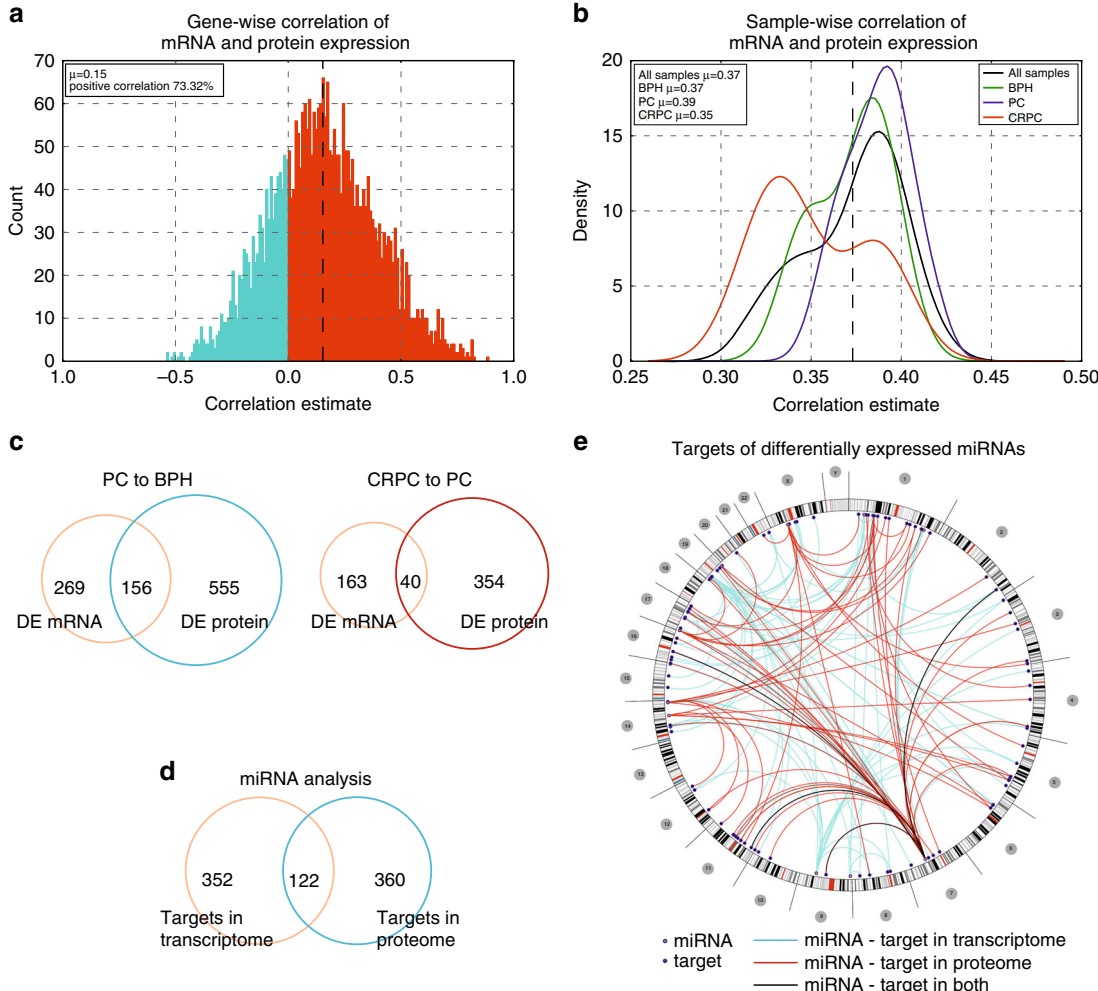

**Fig. 3** Transcriptomic and proteomic data show distinct patterns of expression at the RNA and protein level in prostate cancer. **a** Correlation between mRNA and protein expression of individual genes. The graph shows correlations of all genes identified in proteomic analysis in all samples used in this study. Most of the genes (>73%) show positive correlation between expression of their mRNA and protein. μ is the mean of the correlations. **b** Disease group-wise correlation between mRNA and protein expression of the genes identified in the proteomic analysis shows that in CRPC there is a decreased correlation between mRNA–protein expression pairs compared to primary PC. Compared to all samples (black line) and BPH (green line), the PC samples (blue line) have a higher correlation between their mRNA-protein expression pairs, while CRPC samples (red line) have a lower correlation. μ is the mean of the correlations. **c** Venn diagram showing the numbers differentially expressed (DE) genes in PC vs BPH and CRPC vs PC comparisons identified based on mRNA or protein expression. The numbers of overlapping genes show that only a minority of the differentially expressed genes show expression changes in both mRNA and protein levels. **d** Venn diagram showing the numbers of genes that are negatively correlating with a targeting miRNA based on their expression at the mRNA or protein level. Only a minority of the miRNA targets are identified both at the mRNA and protein level, indicating that correlations at the protein level help to identify mostly a different pool of miRNA targets than correlations at the mRNA level. **e** Circos plot depicting genomic locations of miRNAs and their targets that are both negatively correlating at expression, as well as differentially expressed during prostate cancer progression (CRPC vs PC samples). Outer ring indicates chromosomes and cytobands, with chromosome numbers in the gray circles. Each line in the center maps a prostate cancer-related miRNA-target pair indicated through transcriptomic (blue lines), proteomic (red lines), or both (black lines) analyses. The blue circles mark the genomic location of the miRNAs, and the solid blue dots mark the targets

on a global level, the genomic and epigenomic events that influence mRNA levels are not directly translated to protein expression in prostate cancer.

The effect of altered methylation in prostate cancer on selected genes is, on the other hand, evident also at the proteomics data. There were 140 genes, which were differentially expressed either at mRNA or protein level, with a DMR close by (<10 kb). Within this group, there were several examples of methylation correlating with, and thus likely affecting, mRNA and protein expression. For example, the previously described increased DMR methylation in prostate cancer on genes *ALDH1A2*, *GSTP1*, *GPX3*, and *CYB5R2* correlate with decreased expression of their mRNA and protein according to our data (Supplementary Fig. 3). We further

identified increased promoter DMR methylation in prostate cancer correlating with decreased expression of mRNA and protein expression also on *FBXO2*, *TGFB1I1*, and *TNS1* (Supplementary Fig. 4). Increased gene body methylation in prostate cancer correlating with decreased expression of mRNA and protein expression was identified on *GNAO1*, *LGALS1*, *TNS1*, and *PPAP2B* (Supplementary Fig. 5). Decreased methylation significantly correlating with increased expression was identified for *ENO1*, *SOAT1*, *RPS2*, and *TACSTD2* (Supplementary Fig. 6). Altered DMR methylation found in prostate cancer samples identifies also genes that are less likely to affect directly the outcome of the cancer cells. This is due to either their expression primarily in stromal cells (e.g., *CSRP1*, *CA3*) or the fact that,

despite mRNA expression being affected, the expression level of the protein is not being affected by the differential methylation of the gene (e.g., *CLU, CNTN1*) (Supplementary Fig. 7). Interestingly, we also identified genes whose differentially increased methylation significantly correlated with increased expression in mRNA and/or protein level (*GMDS, MCCC2, MIA3,* and *PYCR1*) (Supplementary Fig. 8).

**Impact of mutations on protein expression**. We identified amino acid altering mutations in expressed genes from the RNA-sequencing data of the samples used in this study, and validated these from the DNA using targeted sequencing (Supplementary Table 4). For all somatic and germline variants, we evaluated the impact of the variant to mRNA and protein expression as described earlier[14]. While somatic mutations had a statistically significant impact to mRNA levels in relation to germline variants (Fisher's exact test, *p*-value = 0.0055, Supplementary Fig. 9), we observed no impact on protein expression levels between somatic and germline mutations or in relation to null distribution estimated from unmutated genes (Supplementary Fig. 9).

To screen for proteins with potential involvement in mutation accrual during prostate cancer development and progression, we assessed correlations of protein expression in relation to mutation

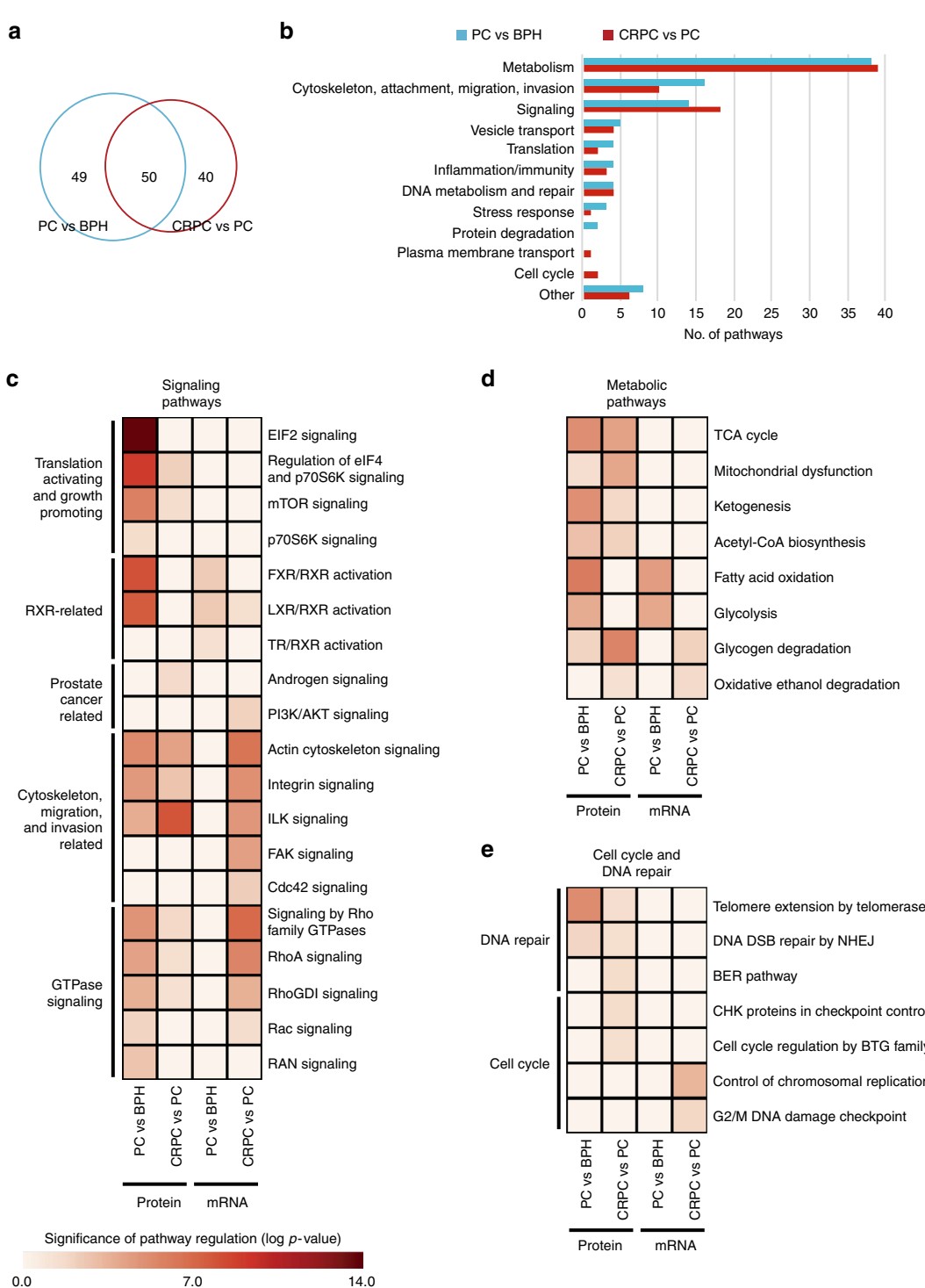

burden of the samples, including the somatic point mutations, copy number alterations, and genomic rearrangements. The two proteins, expression of which correlated best with point mutation burden, were mitochondrial antioxidant regulator PRDX3 (peroxiredoxin 3) and CAD (carbamoyl-phosphate synthetase 2) functioning in de novo synthesis of pyrimidine nucleotides (Supplementary Table 5). For copy number alterations and genomic rearrangements, the best correlating proteins had functions mostly in mitochondria and cytoskeleton. Notably, the strongest negative correlations with the number of rearrangements were with expression of two Talin proteins (TLN2 and TLN1)(Supplementary Table 5).

**Comparison of expression profiles at RNA and protein levels.** The expression levels of most of the proteins identified in our dataset were positively correlated with the expression level of their mRNA, as expected (Fig. 3a). However, when comparing the sample groups, we found that in CRPC, the correlation between individual mRNA-protein pairs was lower in general than in BPH or PC samples (Fig. 3b). We next tested whether similar genes are identified as differentially expressed based on both mRNA and protein expression data. In both PC vs BPH and CRPC vs PC comparisons, only a fraction of the differentially expressed genes were common between the identifications based on transcriptomic and proteomic data, the difference being larger in CRPC vs PC comparison (Fig. 3c). Of the commonly identified genes, 97 and 95% of the differential expressions detected were oriented to the same direction (up or downregulated) in both PC vs BPH and CRPC vs PC comparisons based on mRNA and protein expression data, respectively. According to these results, proteomic and transcriptional data help identify largely different events during prostate cancer development and progression.

Next, we integrated small RNA sequencing data for PC and CRPC samples common between the proteomics and mRNA expression data. MicroRNAs regulate gene expression by binding to mRNA molecules and preventing translation, which leads to decreased target protein expression. miRNA binding to the target can induce degradation of the mRNA, however, also stabilization of the target mRNA has been reported[23]. To study how much of the observed gene expression in prostate cancer is potentially connected to regulation by miRNAs, we studied the pool of differentially expressed genes and their correlating miRNAs. As one miRNA can have several target mRNAs, and one mRNA can be targeted by several miRNAs, we considered individual miRNA-target pairs based on both transcriptome and proteome data, and the predicted or verified miRNA target annotations. Negative correlations between miRNA and differentially expressed targeted mRNAs in CRPC vs PC samples revealed 30 miRNAs and 205 individual miRNA-target pairs (Supplementary

Table 6). Of these, 9 miRNAs were also differentially expressed (Supplementary Table 6). For 34 of the miRNA-target pairs, negative correlation was also found between miRNA and protein expression of the target, indicating a functional impact of miRNA regulation for these particular targets (Supplementary Table 6). To look for the miRNA targets for which the miRNA does not induce mRNA degradation, but effect primarily through inhibition of translation, we searched for negative correlations between miRNA and differentially expressed targeted proteins in the proteome of CRPC vs PC samples. This analysis identified additional 49 miRNAs and 268 individual miRNA-target pairs (Supplementary Table 7). Of these, 8 miRNAs were also differentially expressed (Supplementary Table 7). This pool of miRNA-target pairs represents a resource of novel associations in prostate cancer that have not been visible through previous transcriptome analyses.

To understand the capacity that miRNAs have in regulating prostate cancer progression, we assessed the number of differentially expressed miRNAs and the fraction of the proteome they are collectively able to regulate. There were 95 miRNAs that were differentially expressed between CRPC and PC samples. Assuming negative correlation between a miRNA and its database-predicted or verified target either at the mRNA or protein expression level, the differentially expressed miRNAs in our dataset had the potential to target 16% of the genes in the study. There were 474 and 482 genes according to mRNA and protein expression, respectively, targeted by and negatively correlating with at least one regulating miRNA (Fig. 3d, Supplementary Data 3-4). Of these, only 122 genes were commonly identified (Supplementary Table 8). To look for the miRNA targets which most likely affect prostate cancer progression, we assessed the fraction of the miRNA-regulated genes that were differentially expressed. Of the above miRNA-regulated targets identified based on mRNA expression, 24% ($n = 115$) were differentially expressed between CRPC and PC samples at the mRNA level (Supplementary Fig. 10a, Supplementary Table 9). Similarly, of the regulatory targets identified based on the proteomics data, 45% ($n = 218$) were differentially expressed at the protein level (Supplementary Fig. 10b, Supplementary Table 10). There were 24 genes common between these groups (21% or 11% of the genes identified based on mRNA and protein expression, respectively). A genomic map of the differentially expressed miRNAs and their differentially expressed targets in CRPC vs PC samples based on transcriptomics and proteomics is shown in Fig. 3e. Collectively, these data indicate that by studying the miRNA-target correlations at the protein expression level we were able to identify a significant number of potential regulatory events, which were not identified based on mRNA expression data of clinical prostate cancer samples.

**Fig. 4** Proteomic analysis identifies novel pathways as regulated in PC and CRPC. **a** Venn diagram showing numbers of differentially regulated pathways according to Ingenuity Pathway Analysis in PC vs BPH and CRPC vs PC comparisons. Despite partial overlap, the different disease states have a significant number of pathways specifically regulated. **b** Differentially regulated pathways in **a** according to pathway types. Metabolism is the largest group in both comparisons, with roughly a similar number of pathways differentially regulated. Numbers of most of the other pathway types that are differentially regulated between the disease states vary. **c–e** Examples of signaling pathways found to be differentially regulated according to proteomics (protein) or transcriptomics (mRNA) data in PC vs BPH and CRPC vs PC comparisons. **c** Examples of signaling pathways groups identified as regulated according to proteomic data. Especially translation activating, growth promoting pathways are identified as regulated solely based on proteomic data. RXR-related pathways are identified better by proteomics than transcriptomics to be regulated in PC. Pathways related to cytoskeleton, migration, and invasion, as well as GTPase signaling pathways are identified to be regulated in PC solely by proteomics, although in CRPC they are better identified as regulated by transcriptomics. **d** Metabolic pathways differentially identified as regulated based on proteomic and transcriptomic data include pathways identified as regulated in both PC and CRPC solely based on proteomics (TCA cycle, mitochondrial dysfunction, ketogenesis, acetyl-CoA biosynthesis), and pathways that are equally identified by proteomics and transcriptomics, but are specific for PC (fatty acid oxidation, glycolysis) or CRPC (glycogen degradation, oxidative ethanol degradation). **e** While DNA repair pathways regulated in PC and CRPC were identified based on proteomics only, the regulated cell cycle pathways were altered in CRPC and identified based on either proteomic or transcriptomic data. The color key below panel **c** applies to panels **c**, **d**, and **e**

| Table 1 TCA cycle proteins with altered expression levels in prostate cancer | | | |
|---|---|---|---|
| **Symbol** | **Entrez gene name** | **PC vs BPH** | **CRPC vs PC** |
| ACO2 | aconitase 2 | 3.141 | 0.472 |
| CS | citrate synthase | 1.705 | n.s. |
| FH | fumarate hydratase | 1.598 | n.s. |
| IDH3A | isocitrate dehydrogenase 3 (NAD (+)) alpha | n.s. | 0.653 |
| MDH2 | malate dehydrogenase 2 | 2.167 | 1.912 |
| OGDH | oxoglutarate dehydrogenase | 1.653 | 0.608 |
| SUCLA2 | succinate-CoA ligase ADP-forming beta subunit | 1.909 | n.s. |
| SUCLG1 | succinate-CoA ligase alpha subunit | 2.091 | 0.469 |

Fold changes in protein expression are shown. n.s., not significantly altered

To validate our analysis for miRNA targets detectable both at the mRNA or the protein level, we transfected PC-3 prostate cancer cells with pre-miRNA constructs and assessed the mRNA and protein levels of predicted targets. We selected two representative miRNAs that were differentially expressed to opposite directions during prostate cancer progression, namely miR-22 as downregulated and miR-493 as upregulated in CRPC compared to PC, and verified their successful transfection by TaqMan RT-qPCR (Supplementary Fig. 11a). As positive controls for miRNA targeting at the mRNA level, we performed RT-pPCR on two predicted targets of miR-493 that were identified as negatively correlated based on our analysis at the target transcript level (Supplementary Data 3). Supplementary Fig. 11b shows that, as expected, the mRNA levels of ENDOD1 and GOLM1 are significantly decreased by miR-493 expression. Further, the negatively correlating miRNA-target pairs identified only in the proteomic analysis show decreased protein expression in MS/MS quantification, but no decrease in mRNA levels in RT-qPCR assay, as shown for miRNA-target pairs miR-22—KHRSP1 and miR-493—DNML1 (Supplementary Fig. 11c and d, respectively). These results confirm that our miRNA-target analyses based on the proteomics data have identified miRNA targets that are not identified at the mRNA level.

**Proteomic analysis reveals novel regulated pathways**. To test whether proteomics reveal pathway alterations in prostate cancer that have not previously been found by interrogation of mRNA expression changes, we next performed pathway analysis comparison between mRNA and protein expression data from the same samples. Supplementary Fig. 12a shows that roughly similar numbers of pathways were found significantly regulated based on proteomics and RNA expression data when comparing PC to BPH, and slightly more based on proteomics in CRPC to PC comparison. However, only a minority (16–26%) of the pathways found in each comparison category were common between RNA and proteomics data. These results show that proteomic data is able to reveal pathway regulations not visible at the RNA expression level, especially when comparing CRPC to PC.

We further analyzed which signaling pathways were deregulated during prostate cancer development and progression at the proteomic level. Comparing PC samples to BPH, 99 pathways were found regulated according to Ingenuity Pathway Analysis, while 90 pathways were regulated in CRPC vs PC (Fig. 4a, Supplementary Table 11). Fifty pathways were common between these comparisons. The pathway categories were similar in both comparisons, with metabolic pathways being the most prominent (Fig. 4b). In PC vs BPH, cytoskeleton, attachment, and motility-

related pathways were the second largest group, while in CRPC vs PC it was the signaling pathways. Exclusively in PC vs BPH, there were protein degradation pathways found significantly regulated, while in CRPC vs PC, certain cell cycle pathways were significantly regulated. The pathways common between the PC vs BPH and CRPC vs PC comparisons (Supplementary Table 11) included mostly metabolic pathways, as well as cytoskeleton, attachment and motility-related pathways (62% of the common pathways). It is noteworthy that all significantly regulated DNA metabolism and repair pathways, and most of the vesicle transport pathways, were common between the comparison groups. In contrast, only a few of the regulated signaling pathways were common between the comparison groups, all of which represented Rho GTPase signaling pathways (Supplementary Table 11).

In PC vs BPH, the top significantly regulated pathways included EIF2, eIF4 and p70S6K signaling, as well as FXR/RXR and LXR/RXR activation (Supplementary Table 11, Fig. 4c). While the former pathways promote growth- and survival through alterations in levels of several translation initiation factors and ribosomal proteins, the latter signal to metabolic pathways regulated by farnesoid X receptor (FXR), liver X receptor (LXR), and retinoid X receptor (RXR). When comparing CRPC to PC samples, the most significantly altered pathways during progression of prostate cancer include ILK signaling and glucose metabolism-related pathways (Fig. 4c, d, Supplementary Table 11).

Next, we wanted to further understand the differences in pathway regulation at mRNA and protein levels. Despite being largely different pathways, the biological functions of the pathways most often found by either RNA expression or proteomics were similar, with metabolic, signaling, and cytoskeleton and cell movement-related pathways being the most common (Fig. 4c, d, Supplementary Fig. 12b, Supplementary Fig. 13a,b). Examples of differentially identified signaling pathways are shown in Fig. 4c. Most prominently, the translation-activating and growth-promoting EIF, p70S6, and mTOR signaling, and cytoskeleton-related signaling in PC vs BPH were found solely based on proteomics data. RXR-related signaling in PC vs BPH, as well as several cytoskeleton-related signaling pathways in CRPC vs PC, were found by both transcriptomics and proteomics similarly. Interestingly, GTPase signaling was significantly regulated in PC vs BPH by proteomics and in CRPC vs PC by transcriptomics.

The group of metabolic pathways that was regulated in both PC vs BPH and CRPC vs PC comparisons was extensive (Supplementary Fig. 12b). Despite the relatively low overlap in individual pathways between the comparisons (between disease groups, and between proteomic and transcriptomic data), all the analyses identified pathways from the major groups of energy, amino acid, and lipid metabolism (Supplementary Table 11; examples shown in Fig. 4d, Supplementary Fig. 13b). It is noteworthy that the mitochondria-related metabolic pathways and ketogenesis were identified as differentially regulated only by proteomics, while e.g., glycolytic and glycogen degradation-related pathways were identified by both transcriptomics and proteomics (Fig. 4d). One of the most prominent group of metabolic pathways in prostate cancer were amino acid metabolic pathways (Supplementary Fig. 13b). Interestingly, while different cell cycle regulatory pathways were found regulated by transcriptomic and proteomic data, DNA repair pathways were found solely by proteomics analysis (Fig. 4e). Other interesting groups of differentially identified pathways based on RNA and protein expression were vesicular traffic-related and protein degradation pathways (Supplementary Fig. 13c,d).

**Changes in TCA during prostate cancer evolution**. Based on our analysis, metabolic changes are prominent during both development and progression of prostate cancer. One of the most interesting pathways identified by our proteomic data was the tricarboxylic acid cycle (TCA; also referred to as the citric acid cycle, or the Krebs cycle), which was altered in both PC vs BPH and CRPC vs PC comparisons. This pathway was not found

regulated by RNA expression data, suggesting changes taking place primarily at the protein level. Furthermore, although alterations in certain enzyme activities in TCA have previously been shown to occur during prostate cancer development[24], our proteomics results indicated a previously undescribed, two-step modulation of the TCA cycle. The TCA pathway proteins that were considered regulated by the pathway analysis were mostly

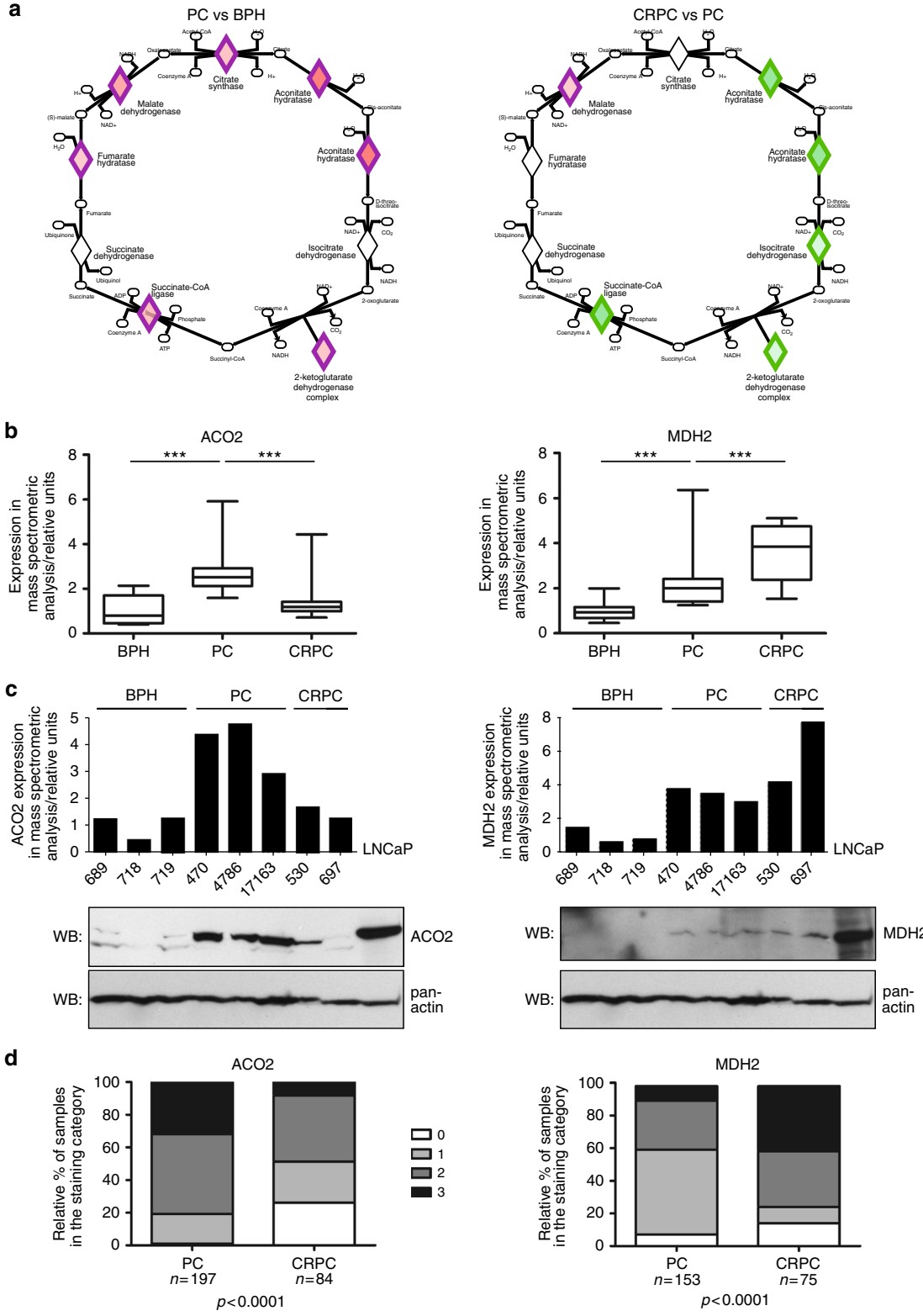

altered to opposite directions in PC vs BPH and CRPC vs PC comparisons: upregulated in PC vs BPH, and downregulated in CRPC vs PC (Table 1, Fig. 5a). An exception was malate dehydrogenase 2 (MDH2), levels of which continued to increase in CRPC (Table 1, Fig. 5a). Comparison of protein and RNA expression of TCA genes[8,23] in all three groups of samples revealed that the TCA proteins are divided into three classes: (1) proteins whose mRNA and protein expression go hand in hand indicating primary regulation by gene expression (CS, FH, IDH3A, IDH2, and SUCLG2), (2) proteins, whose protein levels are not changed (IDH3B, IDH3G), and (3) proteins, that exhibit regulation at the protein level not correlating with mRNA (ACO2, MDH2, OGDH, SUCLA2, and SUCLG1) (Supplementary Fig. 14). From the latter group of proteins, ACO2, OGDH, SUCLA2, and SUCLG1 were all upregulated at the protein level, but not at the mRNA level, in PC vs BPH, while being downregulated in CRPC vs PC either at the mRNA or protein level. Increase in MDH2 protein expression in PC vs BPH did correlate with an increase in mRNA levels, but the increase in CRPC vs PC did not, suggesting posttranslational regulation (Supplementary Fig. 14).

To study more closely the events identified in the TCA cycle, and to validate the results of the proteomics data, we selected two TCA proteins showing significant but different alterations at their protein expression between the prostate cancer sample groups to study further. As a representative of the most common alteration pattern we chose aconitase 2 (ACO2) which showed statistically highly significant ($p < 0.001$, Mann–Whitney test) upregulation of the protein in PC vs BPH, as well as statistically highly significant ($p < 0.001$, Mann–Whitney test) downregulation in CRPC vs PC (Fig. 5b). As a second protein we chose MDH2 exhibiting the deviant behavior amongst the TCA proteins, as it was upregulated statistically significantly both in PC vs BPH ($p < 0.001$; Mann–Whitney test) and further upregulated in CRPC vs PC ($p < 0.05$; Mann–Whitney test) (Fig. 5b). We performed western blotting on these proteins with representative samples of BPH, PC, and CRPC used in the proteomic analysis, and found similar changes than by mass spectrometry (Fig. 5c, Supplementary Fig. 15 and 16), validating the mass spectrometry detection and analysis results.

We performed further validation on the differential regulation of these proteins during prostate cancer progression by immunohistochemical stainings on larger sample sets of clinical PC and CRPC. Grading of the immunohistochemical staining intensity (example staining intensities of grades 0–3 displayed in Supplementary Fig. 17) showed that relative percentage of samples with no or low staining intensities (0–1) of ACO2 increased in CRPC vs PC (Fig. 5d), indicating that the relative levels of ACO2 decreased in CRPC. On the other hand, the relative percentage of samples with higher staining intensities (2–3) of MDH2 increased in CRPC vs PC (Fig. 5d), indicating that the relative levels of MDH2 increased in CRPC. These results

confirm the mass spectrometry results and show that the TCA cycle proteins ACO2 and MDH2 are differentially regulated at the protein level during prostate cancer progression.

We further assessed potential mechanisms that could explain the distinct regulation of MDH2. We found that two miRNAs predicted to target MDH2, namely miR-22 and miR-205, were identified as differentially expressed in our analysis and were negatively correlating with MDH2 protein (Supplementary Data 4) but not mRNA (Supplementary Data 3) levels in the large scale datasets. We transfected PC-3 prostate cancer cells with these miRNAs, and verified the transfection efficiency with TaqMan RT-qPCR analysis (Supplementary Fig. 18a). We detected no significant alterations at MDH2 mRNA levels in RT-qPCR analysis upon elevated expression of the miRNAs (Supplementary Fig. 18b). In contrast, luciferase assay showed statistically significant decrease in reporter production from a MDH2 3′-UTR construct by both miR-22 and miR-205 overexpression (Supplementary Fig. 18c), indicating that these miRNAs are able to directly target MDH2 mRNA. Furthermore, MS/MS quantification showed a substantial decrease in MDH2 protein levels by both miR-22 and miR-205 expression (Supplementary Fig. 18c). These results validated the predictions of miR-22 and miR-205 to directly target MDH2, and identified these miRNAs as prostate cancer-relevant, differentially expressed regulators of the TCA.

## Discussion

We have provided the first extensive proteomic view of prostate cancer development and progression. With over 3000 individual proteins quantified in each of the BPH, PC and CRPC samples analyzed, we described the protein level alterations occurring in clinical prostate cancer, and found several previously undescribed biological events with important implications and potential for future studies. In addition, we provided novel views on the relationship of proteomic, genomic, and transcriptomic changes occurring during castration resistance. The comprehensive view obtained by our integrative analysis underlines the importance of protein level dissection of the molecular mechanisms supporting cancer growth and progression.

Our results showed that neither the altered gene dosages, nor the global methylation changes were translated to the level of the proteome to the same extent as they influence the global RNA expression in CRPC. This suggests that, in the progressed stage, a large proportion of changes in gene copy number and differential DMR methylation are side products of the catastrophic state of cancer cell regulatory systems which are untranslated and thus, subsequently, left without a functional effect at the protein level. Yet, our data confirmed several previously identified regulatory DNA methylation events with associated expression changes occurring in prostate cancer. We also identified several previously

**Fig. 5** TCA cycle is differentially regulated during prostate cancer progression. **a** A schematic view of the TCA cycle protein expression changes in PC vs BPH and CRPC vs PC comparisons according to the Ingenuity Pathway Analysis. Differential expression of TCA enzymes (diamonds) are highlighted in green (downregulation) and red (upregulation). As mostly the same enzymes are involved in both PC and CRPC, the primary mode of expression change is upregulation in PC and downregulation in CRPC. **b** Examples of a typical (ACO2) and a unique (MDH2) TCA protein expression patterns as identified by mass spectrometry proteomics. ACO2 is upregulated in PC compared to BPH, and gets downregulated in CRPC compared to PC. MDH2 protein expression levels increase in PC compared to BPH, and continue to increase in CRPC. Boxplots show interquartiles with mean values, whiskers represent minimum and maximum values. ***$p$-value < 0.001 (Mann–Whitney test). **c** ACO2 and MDH2 protein expression patterns verified in a subset of BPH, PC, and CRPC samples by western blotting. ACO2 and MDH2 protein expression according to the proteomic mass spectrometry analysis (upper panel bar graph) and in corresponding samples according to western blotting (WB; lower panels). Pan-actin is used as a loading control. **d** Change in ACO2 and MDH2 protein expression patterns during progression of prostate cancer verified by immunohistochemistry. Immunohistochemical analysis in clinical tumor samples of PC and CRPC show statistically significantly decreased ACO2 and increased MDH2 staining intensity in CRPC compared to PC and (Chi squared test; 0 = no staining, 1 = weak staining, 2 = intermediate staining, 3 = strong staining)

undescribed protein expression alterations in PC and CRPC associated with differential methylation of DMRs.

We showed that the proteomic profile of prostate cancer is significantly altered during the course of the disease. We identified differentially expressed proteins, potential miRNA regulatory effects, and significantly altered pathway events. The key notion is that these have not been identified through transcriptomic analyses. This supports the view that not all proteins apply to changes at the mRNA level, and underlines the importance of mechanistic studies at the protein level.

Especially intriguing is the group of predicted miRNA-target pairs that we found to have negative correlations between miRNA expression and target protein expression without alterations detected at the target mRNA level. These target mRNAs may be bound by the miRNAs without induced degradation of the target. For each miRNA-target pair, the targeting and relevance for prostate cancer needs to be verified by follow-up experiments. Here, we verified several targets for three example miRNAs that are differentially expressed in CRPC vs PC, and thus may play regulatory roles during prostate cancer progression. Our proteomic pathway analysis identified especially translation-related growth pathways as significantly altered in primary PC compared to BPH samples. In addition, changes in protein degradation pathways were better detected by proteomics than transcriptomics. Thus, protein homeostasis in prostate cancer seems to be regulated primarily at the protein level. In CRPC, the proteome-specific pathway alterations were concentrated on mitochondria-related metabolism and DNA repair. While the glycolytic and long-term energy storage utilization pathways were significantly regulated in prostate cancer at both proteomic and transcriptomic levels, the changes in the core TCA and mitochondrial pathways are evident solely based on the proteomic data. This indicates that posttranscriptional events are taking place in the mitochondria during castration resistance, in order for the cancer cells to ensure survival and propagation under the altered conditions.

As a key finding, we detected two metabolic shifts involving the TCA during prostate cancer development and progression. The changes in TCA enzyme activities during prostate cancer development have been studied earlier, but the second shift occurring during progression to CRPC is previously undescribed. In primary prostate cancer, it is well-established that the normally high tissue citrate levels decrease[24,25]. Costello and Franklin[24] suggested that normal citrate-producing prostate epithelial cells become citrate-oxidizing when they turn malignant. Under this bioenergetic hypothesis, mitochondrial aconitase ACO2 is a key enzyme for the bioenergy transformation[26]. Subsequently, Juang[27] showed that downregulation of mitochondrial aconitase in cultured prostate cancer cells decreases cell proliferation rate. Mitochondrial aconitase gene expression was earlier shown to be regulated by testosterone in prostate epithelial cells in vitro[28], suggesting that in high AR activity tumors ACO2 gene expression could be upregulated. In our gene expression data, ACO2 mRNA levels increase in PC compared to the levels in BPH. However, in CRPC compared to PC, reflecting events during formation of castration resistance and involving increased AR expression, ACO2 mRNA levels are not increased further, and the protein levels decrease. Thus, while our results support previous evidence of upregulation of mitochondrial aconitase levels during development of prostate cancer, progression to CRPC seems to involve primarily posttranslational regulation of the enzyme, reflecting the differences between the first and the second metabolic shift during the course of prostate cancer evolution.

Most of the TCA enzymes are upregulated during the first metabolic shift in prostate cancer, and then either stay upregulated (CS, FH) or are downregulated (e.g. ACO2, OGDH, and SUCLG1)

during the second shift. The exception is MDH2, protein levels of which continue to increase in the second shift during prostate cancer progression. As a mechanism explaining the continued increase in MDH2 protein levels in CRPC, we identified decreased expression of miR-22 and miR-205, miRNAs which were both confirmed to decrease MDH2 protein levels without decreasing the MDH2 mRNA expression. MDH2 is mitochondrial malate dehydrogenase, which is an enzyme that catalyzes the NAD/NADH-dependent, reversible oxidation of malate to oxaloacetate. It has been reported previously that patients with MDH2 overexpression have a significantly shorter period of relapse-free survival after undergoing neoadjuvant combination chemotherapy followed by surgery[29]. Further, stable knockdown of MDH2 via shRNA in prostate cancer cell lines decreased cell proliferation and increased docetaxel sensitivity[29]. Together with our data, these results collectively suggest MDH2 inhibition as a mechanism to target castration resistant tumors. MDH2 druggability has been studied in the context of doxorubicin-induced cardiomyopathy, where the non-specific MDH2 inhibitors mebendazole, thyroxine, and iodine have been found promising[30]. Thus, development of MDH2-specific chemical inhibitors could be of great benefit against progressed prostate cancer, as well as for prevention of cardiotoxicity during chemotherapy.

In conclusion, we identified here several key aspects of prostate cancer biology with the most comprehensive proteomics on primary and progressed prostate cancer samples so far. In addition to increasing our understanding of prostate cancer biology, our study identified several important aspects of prostate cancer signaling and metabolism for future studies.

## Methods

**Samples.** Fresh-frozen tissue specimens from 10 BPH, 17 untreated PC, and 11 CRPC samples were acquired from Tampere University Hospital (Tampere, Finland). PC samples (Supplementary Table 1) were obtained by radical prostatectomy. Mean age at diagnosis was 62.0 years (range: 47.4–71.8) and mean PSA at diagnosis was 9.8 ng/ml (range: 3.5–19.8). Locally recurrent CRPC samples (Supplementary Table 2) were obtained by transurethral resection of the prostate. Samples were snap-frozen and stored in liquid nitrogen. Histological evaluation and Gleason grading were performed by a pathologist based on hematoxylin/eosin-stained slides. All samples contained a minimum of 70% cancerous or hyperplastic cells. The use of clinical material was approved by the ethical committee of the Tampere University Hospital and the National Authority for Medicolegal Affairs. Written informed consent was obtained from the subjects.

**Chemicals and materials.** Acetonitrile (ACN), formic acid (FA), water (UHPLC-MS grade), triethyl ammonium bicarbonate buffer (TEAB), sodium dodecyl sulfate (SDS), iodoacetamide (IAA), trifluoro acetic acid (TFA), ammonium bicarbonate (ABC), tris-(2-carboxyethyl)phosphine (TCEP), urea and pellet pestles were all purchased from Sigma Aldrich (St. Louis, MO, USA). RIPA lysis buffer, protease inhibitor cocktail (Halt™) and sample clean up tips (C18) were from Thermo Fisher Scientific (San Jose, CA, USA). Bio-Rad DC™ protein assay kit and bovine serum albumin standard were purchased from Bio-Rad (Hercules, CA, USA) and 30 kDa MWCO centrifugal devices from PALL (Port Washington, NY, USA). TPCK-treated trypsin was from AB Sciex (Framingham, MA, USA). HRM Calibration Kit was purchased from Biognosys AG (Zurich, Switzerland).

**Protein extraction from tissue samples and enzymatic digestion.** Five 5 μm slices were cut from fresh-frozen tissue samples. Tissues were homogenized with polypropylene pellet pestle in ice-cold RIPA lysis buffer containing Halt protease inhibitor. The disrupted tissues were subjected to sonication for 5 min followed by a 30 min incubation on ice. After incubation, lysates were centrifuged to remove any remaining cell debris (16,000 xg, 20 min, +4 °C). Total protein concentration of the samples was measured with Bio-Rad DC protein assay. Mean amount of protein recovered from frozen tissues was 91.5 ± 67.3 μg (SD). From 9 to 50 μg of protein was precipitated with acetone (−20 °C) overnight. The protein amounts were selected based on our previous testing of suitable injection volume of 5 μg total protein in 2 μl volume in SWATH. Precipitated proteins were centrifuged, supernatant was decanted, and samples were allowed to dry for 5 min. Proteins were dissolved in 0.05 M ABC with 2% SDS and reduced by 0.05 M TCEP. After 60 min of incubation at + 60 °C, samples were transferred into 30 kDa molecular weight cut-off centrifugal filters and flushed twice with 8 M urea in 0.05 M Tris-HCl. Cysteine residue blocking was carried out by 0.05 M IAA in 0.5 M Tris-HCl at room temperature in the dark. Samples were repeatedly flushed with 8 M urea and

0.05 M ABC to remove urea prior to digestion with trypsin for 16 h at + 37 °C at a trypsin-to-protein ratio of 1:25. Digests were collected by rinsing the centrifugal devices with 0.1 M TEAB followed by 0.5 M NaCl and dried in a speed vacuum concentrator. Samples were dissolved in 0.1% TFA and desalted with C18 tips. Sample clean-up and desalting was performed with Pierce C18 tips according to manufacturer's instructions. Samples were dried in speed vacuum concentrator and stored at −20 °C until reconstituted in loading solution (5% ACN, 0.1% FA) at equal concentrations. HRM peptide mix was added to each sample before NanoRPLC-MSTOF SWATH analysis.

**NanoRPLC-MSTOF for discovery proteomics.** Digested peptides were analyzed by Nano-RPLC-MSTOF instrumentation using Eksigent 425 NanoLC coupled to high speed TripleTOF™ 5600 + mass spectrometer (Ab Sciex, Concord, Canada). A capillary RP-LC column (cHiPLC® ChromXP C18-CL, 3 µm particle size, 120 Å, 75 µm i.d × 15 cm, Eksigent Concord, Canada) was used for LC separation of peptides. Samples were first loaded into trap column (cHiPLC® ChromXP C18-CL, 3 µm particle size, 120 Å, 75 µm i.d × 5 mm) from autosampler and flushed for 10 min at 2 µl/min (2% ACN, 0.1% FA). The flush system was then switched to line with analytical column and gradient alution. All samples were analyzed with 120 min 6 step gradient using eluent A: 0.1% FA in 1% ACN and eluent B: 0.1% FA in ACN (eluent B from 5 to 7% over 2 min, 7 to 24% over 55 min, 24 to 40% over 29 min, 40 to 60% over 6 min, 60 to 90% over 2 min and kept at 90% for 15 min, 90 to 5% over 0.1 min and kept at 5% for 13 min) at 300 nl/min.

In order to perform SWATH-MS quantification, we first generated a spectral identification library with 57 different samples (prostate tissue and cancer cell line samples). Key parameters for MSTOF mass spectrometer in SWATH ID library analysis were: ion spray voltage floating (ISVF) 2300 V, curtain gas (CUR) 30, interface heater temperature (IHT)+125 °C, ion source gas 1 13, declustering potential (DP) 100 V. All methods were run by Analyst TF 1.5 software (Ab Sciex, USA). For IDA parameters, 0.25 s MS survey scan in the mass range 350–1250 mz were followed by 60 MS/MS scans in the mass range of 100–1500 Da (total cycle time 3.302 s). Switching criteria were set to ions greater than mass to charge ratio (m/z) 350 and smaller than 1250 (m/z) with charge state 2–5 and an abundance threshold of more than 120 counts. Former target ions were excluded for 12 s. Information dependent acquisition (IDA) rolling collision energy (CE) parameters script was used for automatically controlling CE. SWATH quantification analysis parameters were the same as for spectral identification library analyses, with the following exceptions: cycle time 3.332 s and MS parameters set to 15 Da windows with 1 Da overlap between mass range 350–1250 Da followed by 40 MS/MS scans in the mass range of 350–1250 Da.

**Mass spectrometric data analysis.** SWATH library analysis were performed with Protein pilot software version 4.7 (Ab Sciex, Canada) which was used to analyze MS/MS data and searched against the UniprotKB/Swiss-prot database for protein identification. Settings in the Paragon search algorithm in Protein pilot were configured as follows. Sample type: identification, Cys-alkylation: MMTS, Digestion: Trypsin, Instrument: TripleTOF 5600 + , Search effort: thorough ID. False discovery rate (FDR) analysis was performed in the Protein pilot and FDR < 1% was set for protein identification. The data from all the identification runs were combined as a batch and used for library creation for SWATH relative quantification.

For quantification we used PeakView® software 2.0 with SWATH-plug in to assign the correct peaks to correct peptides in the library. Two replicate MS analyses were done from each sample. iRT peptides (Biognosys, Switzerland) was used for retention time calibration with PeakView. 1–15 specified peptides per protein were selected to be used in SWATH quantification. Peptide peak areas were extracted and filtered to remove all peptides, which do not have a single measurement with an FDR <1% across all measurements. The SWATH-MS data exhibited excellent quality and reliability with $p$-value < 0.05 in 98.6% of replicate MS analyses (permutation tests, Spearman's rank correlation) and mean interclass correlation (ICC) coefficient of 0.98.

**Statistical analysis of proteomics data.** Data processing included $\log_2$-transformation and quantile normalization. The quality of the replicate MS analyses was analyzed by calculating the intraclass correlation (ICC) and Spearman's rank correlation was used to generate $p$-values in permutation tests ($n = 1000$ permutations/replicate MS analyses). Further analysis was performed on the mean values of the replicate MS analyses. Wilcoxon rank sum test was implemented to analyze the differences between sample types. Benjamini and Hochberg adjustment were applied to all initial $p$-values, where applicable, to account for the multiple testing issues. R software version 3.2.3 (R Core Team. Foundation for Statistical Computing, Vienna, Austria) was used to analyze data. Ingenuity Pathway Analysis (IPA, QIAGEN Redwood City, USA) was used to conduct pathway analysis and identify proteins connected to pathways of interest. Protein grouping and classification was performed by using PANTHER Classification System[31].

**Analysis of differentially expressed mRNA and protein.** Common samples (Supplementary Table 3) between the proteomic analysis performed here and previously described mRNA expression (RNA sequencing) data[7] were used to

extract common genes ($n = 3310$) between the protein and mRNA data. mRNAs and proteins were considered differentially expressed across different comparisons (BPH vs PC, PC vs CRPC) if absolute median ratio of two conditions was greater than 1.5 with a $p$-value < 0.05 (Benjamini–Hochberg adjusted $p$-value of a non-parametric Wilcoxon test).

**Association between protein expression and gene copy number.** The Spearman's rank correlation was calculated for each sample between gene level DNA copy numbers[7] and mRNA expressions, or copy numbers and protein expressions (Supplementary Table 3). Using the correlation values probability density functions (PDFs) for each correlation value sets were estimated using kernel density estimation with Gaussian kernels and Scott's rule for bandwidth determination. The estimated PDFs were then plotted and supplemented by rug plots of the exact correlation values. To estimate background distributions, the Spearman's rank correlations of each sample with all other samples were calculated between copy numbers and mRNA expressions or copy numbers and protein expressions. Using the correlation values PDFs for each correlation value set were estimated as detailed above.

**Association between protein expression and DNA methylation.** Based on MeDIP-sequencing data[7] we identified 751 differentially methylated regions (DMRs) within 10 kb from TSS of 557 unique genes with available expression values for RNA expression and protein expression (Supplementary Table 3). Subsequently, the Spearman's rank correlations were calculated for each sample between DMR normalized fragment counts and mRNA expressions or DMR normalized fragment counts and protein expressions. Kernel density estimation was used for visualization of the correlation values as described above. Background distributions were calculated in the same manner as explained earlier. Furthermore, we identified 2773 genes common between mRNA and protein expression datasets where their absolute distance to a nearby DMR was <250 kb. Of these, 745 genes were showing absolute correlation >0.3 between their gene expression and nearby DMRs. 140 out of 745 genes were differentially expressed both at mRNA and protein level. Finally only 79 of these genes had absolute distance ≤10 kb from 117 DMRs (these were used for scatter plots).

**Structural variation analysis.** To identify rearrangements whole genome sequencing reads were aligned against the GRCh37 reference genome using Bowtie-2.0.0-beta7[32]. An in-house structural variant calling software called Breakfast (https://github.com/annalam/pypette) was then used to identify paired end reads where the mates aligned discordantly. A paired alignment was considered discordant if both mates aligned to the genome but aligned to separate chromosomes or >100 kb apart. Mates with an alignment quality phred value < 20 were discarded from analysis. Next, individual mates that did not initially align to the reference genome were split into 25 bp anchors. The 25 bp anchor pairs were then realigned and searched for discordant alignments using the same criteria as with paired end reads. The full 90 bp sequences corresponding to discordant anchor pairs were compared against the reference genome to identify exact breakpoints and to analyze for sequence homologies. A discordant anchor pair was discarded if the sequence homology between the read and one of the breakpoint flanking sequences was above 70% for the nucleotides matching with the discordant anchor. The exact breakpoint was determined by selecting the breakpoint associated with the lowest amount of nucleotide mismatches. After identifying discordant pairs from paired end and split reads, the discordant pairs were reoriented so as to always have the pair with the lower chromosome or coordinate first. Discordant pairs were then clustered using a sliding window approach. A cluster of discordant pairs was accepted as a putative structural variant if it contained at least one paired end read and one split read indicating the structural variant. To filter out false positives, structural variants were also called in BPH samples, and all genomic regions within 1 kb of a breakpoint identified in a BPH sample were blacklisted.

**Point mutation impact analysis.** Somatic and germline point mutations[22] in each sample were used to find their impact on the expression level of the genes harboring the mutations as described earlier[14]. Null distribution was generated by comparing expression of randomly selected unmutated sample to other unmutated samples.

**Association between protein expression and mutation burden.** Number of somatic point mutations[22], rearrangements (as described above), and chromosomal instability (CIN) in each sample across common samples between protein and mRNA expression data (Supplementary Table 3) were used to find their association (Spearman's rank correlation) with individual genes in the protein dataset. CIN was calculated as the mean of integer copy numbers assigned to non-overlapping blocks of size 500 bp spanning across the entire genome.

**Association between protein and miRNA expression.** miRNA expression data (small RNA sequencing)[7] were used to extract miRNAs with negative correlation with their targets using the common samples (Supplementary Table 3). Predicted

targets of miRNAs were downloaded from miRWalk 2.0 database[33] using the following parameter values: Input parameters Promoter 2 kb, 3′ UTR, minimum seed length 7 and/or p-value 0.05. We considered mRNA to be a target for miRNA if targeting was predicted by 2/3 of the databases miRanda, PICTAR2, and Targetscan[34–36]. Differentially expressed miRNAs were defined as having an absolute median ratio between two conditions >1.5, and the Benjamini-Hochberg adjusted p-value of a non-parametric Wilcoxon test <0.05. miRNAs were considered unexpressed if all samples had read count below 8 and they were excluded from differential expression analysis. Spearman's rank correlations were calculated between the miRNA expression and the expression of its predicted targets with a threshold for negative correlation $< = -0.50$. For enrichment analysis, hypergeometric test was used to test statistical significance (p-value < 0.05) of the number of negatively correlating predicted targets of a miRNA. miRNA—target associations were visualized as circos plot using POMO[37].

**Transfections of pre-miRNA**. PC-3 cells (ATCC, Rockville, MD, USA) were cultured under the recommended conditions and reverse transfected with 10 nM non-targeting control (miR-control) or pre-micro-RNA constructs (Applied Biosystems/Ambion, Austin, TX, USA) using INTERFERin transfection reagent (Polyplus Transfection SA, Illkirch, France) according to manufacturer's instructions. Cells were incubated for 48 or 72 h before collection for RNA or protein samples, respectively.

**RNA extraction and RT-qPCR**. RNA was extracted using TriReagent® (Sigma-Aldrich) according to manufacturer's instructions. Quantitative RT-PCR for miRNAs was performed using TaqMan microRNA Assay (Applied Biosystems, Foster City, CA, USA) according to the manufacturer's recommendations. RNU6B was used as a reference gene. Quantitative RT-PCR for mRNAs was performed using Maxima SYBR Green (Fermentas Inc., Burlington, Ontario, Canada) from cDNA made using Maxima RT reverse transcriptase (ThermoFischer Scientific Inc.). TBP was used as a reference gene. qPCR reactions were performed with the CFX96 q-RT-PCR detection system (Bio-Rad Laboratories Inc., Hercules, CA, USA).

**Luciferase reporter assay**. PC-3 cells were reverse transfected with 10 nM non-targeting control (miR-control) or pre-micro-RNA constructs (Applied Biosystems/Ambion, Austin, TX, U.S.A.), and MDH2-3′-UTR in pEZX-MT05-GLuc-SEAP luciferase reporter plasmid (GeneCopoeia, Rockville, MD, USA; 10 ng/well) in 96 well plates using jetPRIME transfection reagent (Polyplus Transfection SA, Illkirch, France) according to manufacturer's instructions. Cells were incubated for 24 h before the medium was collected for analysis of secreted Gaussia luciferase (GLuc) and secreted alkaline phosphatase (SEAP) activities with Secrete-Pair™ Dual Luminescence Assay Kit (GeneCopoeia) according to manufacturer's instructions.

**MicroLC-MSTRAP for targeted protein validation analysis**. Proteins for targeted MS/MS analysis were selected based on their expression in discovery analysis. Peptides for each protein were selected based on their specificity, intensity (based on SWATH-MS analysis), amino acid composition, and water solubility in the tissue samples. All peptides with methionine or modifications or missing cleavage sites were disqualified. For each selected peptide, an isotopically labeled standard peptide (AQUA-peptides, Sigma-Aldrich) was used to confirm the identification. For each protein in the analysis, two peptides for targeted MS analysis were selected, and each peptide analysis was confirmed using 3 fragment ions. The peptides, fragment ions, and corresponding isotopical standards for each protein are represented in Supplementary Table 12.

Cell lysis, protein measurements, and tryptic digestion were performed as before. TEAB-solution supplemented with 20 fg of each targeted peptide isotope per 1 μg of total protein in the sample was used to flush the digested peptides of the membrane. Sample cleanup was performed as before. 1 μg of cleaned samplewas used for MicroLC-MSTrap analysis. Analysis was performed using Sciex 6500 + MSTrap coupled with Eksigent NanoLC 425 with 1–10 μl/min microLC flow cell. MicroLC utilized a 42 min 6 step gradient using eluent A: 0.1% FA in MQ and eluent B: 0.1% FA in ACN (eluent B from 10 to 30% over 22 min, 30–50% over 8 min, 50–80% over 2 min, kept at 80% for 5 min, 80–10% over 0.2 min and kept at 10% for 5 min, at 5 μl/min. MSTrap settings were as follows; Curtain gas: 30, Spray voltage: 5300, Collision gas: medium, Temperature: 150 °C, Ion source gas 1: 20, Ion source gas 2: 20, were set the same for all peptides. Collision energy was specifically set to 40 for OLA1 peptide IPAFLNVVDIAGLVK and to its respective isotope standard and to 30 for all others. Results were normalized against their representative isotopically labeled standard peptide and then compared between samples. Standard deviation for each peptide in the analysis method was calculated using isotope labeled peptide standards. Relative standard deviation for all the peptides are under 10%.

**Western blotting**. LNCaP cells (ATCC) were cultured under the recommended conditions. Cells and sections of frozen tissue were lysed in Triton-X lysis buffer containing 50 mM Tris-HCl pH 7.5, 150 mM NaCl, 0,5% Triton x-100, 1 mM PMSF, 1 mM DTT and 1× complete protease inhibitor cocktail (Roche Inc., Mannheim, Germany), after which the lysates were sonicated four times for 30 s at medium power with Bioruptor equipment (Diagenode Inc., Liège, Belgium), and cellular debris was removed by centrifugation. Proteins were separated by polyacrylamide gel electrophoresis (SDS-PAGE) and transferred to PVDF membrane (Immobilon-P; Millipore Inc., Billerica, Massachusetts, USA). Primary antibodies against ACO2 (HPA001097; Sigma-Aldrich, St. Louis, MO, U.S.A.; dilution 1:1000), MDH2 (HPA019714; Sigma-Aldrich; 1:1000), and pan-actin (ACTN05; NeoMarkers, Portsmouth, NH, USA; 1:1000) were used and detected using anti-rabbit HRP-conjugated antibody produced in swine (1:5000, DAKO Inc., Denmark) or by anti-mouse HRP-conjugated antibody produced in rabbit (1:5000, DAKO Inc., Denmark) and Western blotting luminol reagent (Santa Cruz Inc., Santa Cruz, California, USA) with autoradiography. Original scans including molecular weight information for the western blots are presented in Supplementary Fig. 19.

**Immunohistochemistry**. Formalin-fixed, paraffin-embedded tumor microarrays of PC and CRPC samples[38] were used. Sections were deparaffinized and antigen retrieval was performed by using Tris-EDTA buffer 0.05% Tween-20 (pH 9) at + 98 °C for 15 min. The staining was performed by Lab Vision Autostainer (ThermoFischer Scientific Inc., Waltham, MA, USA). Primary antibodies (as above) and secondary antibody (N-Histofine® Simple Stain MAX PO; Nichirei, Tokyo, Japan) were used. ImmPACT DAB (Vector Laboratories, Burlingame, CA, USA) was used as a chromogen. The sections were counterstained with hematoxylin and mounted with DPX mounting medium (Sigma-Aldrich). Scoring of staining intensity on tumor areas was performed on a 0–3 scale (Supplementary Fig. 17), and the difference in score distributions between PC and CRPC groups was statistically assessed with Chi squared test.

**Data availability**. Mass Spectrometry data has been deposited to Peptide Atlas repository under dataset identifier PASS01126. Deep sequencing data has been deposited to European Genome-Phenome Archive under accession number EGAS00001000526.

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

## Acknowledgements

We thank Paula Kosonen, Riina Kylätie, Artturi Lassila, Katja Liljeström, Saara Lähde-korpi, Päivi Martikainen, and Marika Vähä- Jaakkola for their technical assistance and M.D. Teemu Tolonen for his professional assistance. This work was supported by grants from the Academy of Finland (Project Nos. 269474 MN, 127187 TV), Sigrid Juselius Foundation (M.N., T.V.), Cancer Society of Finland (M.N., T.V.), Competitive State Research Financing of the Expert Responsibility area of Tampere University Hospital (TV, MN, TT), TEKES (Project No. 66/31/2012 U.A., R.B., A.J., J.N., H.U.), and Elsemay Björn Fund (U.A., A.J., J.N., H.U.). We would also like to acknowledge CSC—IT Center for Science Ltd. (https://www.csc.fi/csc) for providing the computational resources.

## Author contributions

U.A., R.B., M.N., H.U., and T.V. conceived and supervised the study. All authors designed and discussed experiments. A.J. carried out the mass spectrometry and SWATH analysis. A.J. and J.N. performed proteomics data analysis. E.A. performed integrative bioinformatics analyses. M.A. and K.W. performed mutation analyses. L.L. and J.N. performed pathway analysis. L.L. carried out western, immunohistochemical, and cellular analyses, and prepared the manuscript. All authors contributed to writing of the manuscript, as well as reviewed and accepted the manuscript.

## Additional information

**Competing interests:** The authors declare no competing interests.

