## [Peer Review File(PDF 370 kb) · Nature Communications]

Reviewers' comments:

Reviewer #1 (Remarks to the Author):

This manuscript describes quantitative proteomic analysis of 37 prostate tissue samples from three groups of patients using SWATH mass spectrometry and reports expression of 3394 proteins in all samples. The authors then compare the proteomic data with published genomic and transcriptomic data from some of the samples, and identified correlation between miRNA and some proteins. By comparing proteomics data among patient groups, the authors proposed the TCA cycle as a potentially critical process during prostate cancer progression. Two TCA proteins (ACO2 and MDH2) are validated using WB, and further validated using larger sample sets using IHC.

The miRNA-target pairs analysis is interesting however the results are not extensively discussed. The authors are advised to discuss and further validate at least a few of the observed pairs before claiming it is a useful resource and clinically significant result.

The quality of the proteomics data is not convincing unless the following issues are properly addressed.

1, it is unclear whether and how the proteomics experiments were performed with replicates. The authors are requested to provide the protein and peptide quantification data matrices in the supplementary tables, and to demonstrate the quality metrics of the protein matrix, including FDR estimates.

2, the authors are requested to provide more details in Supplementary table 4 to show what samples are common between data sets.

3, Fig5c: the authors should clearly state which representative samples were selected for WB validation. The quality of WB results is not optimal and convincing.

Reviewer #2 (Remarks to the Author):

The manuscript describes mass spectrometry-based global analysis of proteome associated with BPH, localized PCa and locally recurrent lymph node metastasis. Authors have used the SWATH technique to profile about 4000 and odd proteins. In addition, for most of the samples (except BPH) data on matched analysis of genomic alterations, transcriptome, methylome and miRNA that were previously carried out have been used for analysis. The authors compare data from the different omics compartments and describe the degree of correlation (positive vs negative vs none). The concluding part of the manuscript deals with aconitase 2 and malate dehydrogenase that authors select for validation studies (immunoblot and tissue staining) and demonstrate that ACO2 is elevated in PCa vs benign but downregulated in lymph node met vs PCa, while Malate dehydrogenase is progressively elevated from BPH to PCa to lymph node met.

The strength of this study is the global analysis of 4000 proteins in prostate cancer associated tissues and availability of matched OMICs data to cross compare and possibly integrate

A number of weakness exist. First and foremost is their definition of CRPC which in clinical terms is used to designate patients who exhibit biochemical recurrence and are treated with second generation antiandrogens and become refractory to these regimens. Samples used here are non-regional lymph node samples. Lymph node involvement at the time of primary resection is an indicator of poor prognosis but the specimens are not CRPC. This is a major caveat.

Secondly, the concept of data integration is limited to comparison of relative expression levels of analytes in the various compartments. It is well known that the degree of correlation between transcript and protein for example is low and part of this is due to existence of PTM's. This issue is completely ignored in the study. The entire manuscript deals with numerous comparisons with lack

of rationale as to why each of these are performed and no mention about the take home message. It is not clear what is the rationale looking at common set of analytes between BPH and PCa vs PCa and lymph node met. BPH is not a precursor for PCa. The authors should have used benign adjacent tissue as a control to look at PCa. Also it is unclear what is the contribution of different cell types to the various results. Example in the comparison between BPH and PCa, GTPase is regulated solely at the proteome level but the same signaling process is regulated by transcriptome when comparing CRPC vs PCa. Why is this the case? Is this due to heterogeneity in cell population? Also when making correlations, authors do not consider the fact that mass spectrometry-based proteomics is inherently affected by missingness of the data in which case many of the proteins corresponding to transcripts seen in the mRNA dataset for example may be missing. How is this accounted for? None of the proteomics findings have been orthogonally validated except ACO2 and MDH2. Here again it is unclear what is the role of increased ACO2 in PCa. It is well known that zinc transporters regulate ACO2 function and that Zn is an inhibitor of ACO2 activity. What does increase in ACO2 mean? What does progressive change in MDH2 mean? None of these have been addressed.

In summary, there is no evidence presented to state that proteomics data in this case has uncovered driver events in prostate cancer progression as stated by the authors in the abstract.

Response to reviewers:

We thank the reviewers for the insightful comments, and have revised the manuscript accordingly. We have marked additions and changes in the manuscript text in blue colour.

Reviewer #1 (Remarks to the Author):

This manuscript describes quantitative proteomic analysis of 37 prostate tissue samples from three groups of patients using SWATH mass spectrometry and reports expression of 3394 proteins in all samples. The authors then compare the proteomic data with published genomic and transcriptomic data from some of the samples, and identified correlation between miRNA and some proteins. By comparing proteomics data among patient groups, the authors proposed the TCA cycle as a potentially critical process during prostate cancer progression. Two TCA proteins (ACO2 and MDH2) are validated using WB, and further validated using larger sample sets using IHC.

The miRNA-target pairs analysis is interesting however the results are not extensively discussed. The authors are advised to discuss and further validate at least a few of the observed pairs before claiming it is a useful resource and clinically significant result.

We thank the reviewer for the suggestion and have now performed additional analysis to validate miRNA-target pairs. The data is presented in two new supplementary figures. In supplementary figure 11, we show targets for two representative miRNAs that are differentially expressed to opposite directions in CRPC compared to PC. We show verification for both targets that are decreased by miRNA overexpression at the mRNA level, as well as targets whose mRNA levels remain unaltered, but protein levels decrease as expected according to the *in silico* analysis. Furthermore, in Supplementary figure 18 we verify two predicted, differentially expressed miRNAs to target MDH2 with no alteration at the mRNA level, but detectable effect at the protein level and inhibitory ability of the specific miRNAs in MDH2-3-UTR-Luciferase assay. These results confirm direct targeting of the MDH2 by the two miRNAs.

We have also added a paragraph of discussion on the miRNA-target regulation, and discussion on the MDH2 targeting miRNAs to the manuscript text.

The quality of the proteomics data is not convincing unless the following issues are properly addressed.

1, it is unclear whether and how the proteomics experiments were performed with replicates.

Each sample in the proteomics has been analyzed in two replicate MS analyses. We have added this information to the manuscript text in materials and methods section.

Raw data has also been transmitted to Peptide Atlas. The data can currently be viewed with a password that is provided here for review purposes. The data will be made publicly available at the publishing date. The data can be viewed here:

<http://www.peptideatlas.org/PASS/PASS01126> (password:PCnature)

2, the authors are requested to provide more details in Supplementary table 4 to show what samples are common between data sets.

We have included the sample-wise overlaps between the different analyses as a Supplementary Table 4b.

3, Fig5c: the authors should clearly state which representative samples were selected for WB validation. The quality of WB results is not optimal and convincing.

The sample numbers have now been added to Figure 5c. To provide further supportive evidence of protein level changes, we present the chromatograms of the mass spectrometry analysis for ACO2 and MDH2 in Supplementary figures 15 and 16, exhibiting the clear difference in expression of these proteins between the sample groups.

In order to obtain more convincing evidence in terms of protein level quantification, the additional experiments in the revised manuscript (concerning the miRNA targets) have been performed with targeted MS/MS analysis (targeted mass spectrometry method for MicroLC-MSTrap using stable isotope labeled peptide standards for absolute quantification).

Reviewer #2 (Remarks to the Author):

The manuscript describes mass spectrometry-based global analysis of proteome associated with BPH, localized PCa and locally recurrent lymph node metastasis. Authors have used the SWATH technique to profile about 4000 and odd proteins. In addition, for most of the samples (except BPH) data on matched analysis of genomic alterations, transcriptome, methylome and miRNA that were previously carried out have been used for analysis. The authors compare data from the different omics compartments and describe the degree of correlation (positive vs negative vs none). The concluding part of the manuscript deals with aconitase 2 and malate dehydrogenase that authors select for validation studies (immunoblot and tissue staining) and demonstrate that ACO2 is elevated in PCa vs benign but downregulated in lymph node met vs PCa, while Malate dehydrogenase is progressively elevated from BPH to PCa to lymph node met.

The strength of this study is the global analysis of 4000 proteins in prostate cancer associated tissues and availability of matched OMICs data to cross compare and possibly integrate

A number of weakness exist. First and foremost is their definition of CRPC which in clinical terms is used to designate patients who exhibit biochemical recurrence and are treated with second generation antiandrogens and become refractory to these regimens. Samples used here are non-

regional lymph node samples. Lymph node involvement at the time of primary resection is an indicator of poor prognosis but the specimens are not CRPC. This is a major caveat.

The samples used in this study were freshly frozen tissue specimens from BPH obtained from cystoprostatectomies or transurethral resections (TURP), untreated PC obtained from prostatectomies, and CRPC obtained from TURPs. The CRPC samples came from patients that had been treated either by castration and/or antiandrogens and experienced urethral obstruction (ie. local recurrence) during the treatment. No lymph node samples were used in this study.

Secondly, the concept of data integration is limited to comparison of relative expression levels of analytes in the various compartments. It is well known that the degree of correlation between transcript and protein for example is low and part of this is due to existence of PTM's. This issue is completely ignored in the study.

We agree with the reviewer that the PTMs are a biologically very relevant and interesting issue. Regulation by PTMs may explain the discrepancy in transcript and protein abundance for several proteins as the main mechanism, or one of several mechanisms. To address PTMs on a large scale is a highly important aspect of protein regulation, and has been addressed in prostate cancer samples recently by Drake et al. (2013,2016). In this study, we focused on protein abundance and the correlations of the proteomic data to genetic, epigenetic and transcriptomic data. In the future, it would be extremely interesting to obtain quantitative, large-scale data on PTMs from our sample set. We thank the reviewer for raising this important point, and have included discussion of this topic to the manuscript.

The entire manuscript deals with numerous comparisons with lack of rationale as to why each of these are performed and no mention about the take home message. It is not clear what is the rationale looking at common set of analytes between BPH and PCa vs PCa and lymph node met. BPH is not a precursor for PCa. The authors should have used benign adjacent tissue as a control to look at PCa.

We agree with the reviewer that BPH is not a precursor of PCa. As fully normal prostate tissue is difficult to obtain especially from elderly men, the benign tissues are often used instead as the baseline reference for PC samples. We agree with the reviewer that benign adjacent tissue would be a good option to represent the baseline of prostate tissue. As benign adjacent tissue was not available from these samples, we used the other commonly used and widely accepted reference of BPH.

As for the lymph node mets, no such samples were used in this study. The comparison between PC and BPH represents a comparison between primary cancer and benign prostate tissue, which is able to reveal at least partly the alterations taking place during initial cancer formation. The comparison between CRPC and PC represents comparison between untreated and recurrent cancer, revealing alterations occurring during development of treatment resistance and cancer progression.

As to the rationale for the different comparisons, we would emphasize the novelty of our data and approach. Large-scale proteomics data has not been previously reported from both primary and advanced

disease in prostate cancer. In addition to novel information of the proteomic output of the previously reported genetic and RNA expression data, we show for the first time how the increased aberration load in CRPC is not similar at the level of the proteins, showing how genetic or RNA expression aberrations cannot be solely used to interpret the molecular state of cancer cells. Although the information can still be complemented by other large-scale analyses, such as metabolomics and PTMs, as kindly pointed out by the reviewer, our results provide a whole new level of knowledge to the molecular events in prostate cancer. In addition, our dataset provides an enormous resource for the cancer research community, as the datasets will be publically available for future studies.

We have clarified sample comparison set-up and motivations in the manuscript text.

Also it is unclear what is the contribution of different cell types to the various results.

This is a relevant question that concerns most studies using clinical tissue material, and what needs to be addressed in more detailed studies of tissue heterogeneity and microenvironment. We have secured that our samples contain over 70% of cancerous or hyperplastic cells, in order to make sure that most of the signals we detect originate from the desired epithelial compartment. This information is provided in the materials section and now also raised in the discussion.

Example in the comparison between BPH and PCa, GTPase is regulated solely at the proteome level but the same signaling process is regulated by transcriptome when comparing CRPC vs PCa. Why is this the case? Is this due to heterogeneity in cell population?

Our analysis points out the levels of expression that contain significant differences between the sample groups. As described above, the hyperplastic and tumor cell compartments comprise the majority of our samples, but it remains a possibility that other cell types may contribute to the results. While the large scale analysis presented here is aiming to identify pathways worth focusing future efforts on, the detailed conclusions on any altered pathways need to be based on verification experiments by supportive methods, such as IHC for detection of cells of signal origin, in similar manner that we have performed here for the TCA cycle. We have added the point of cellular heterogeneity contributing to the results to the discussion.

Also when making correlations, authors donot consider the fact that mass spectrometry-based proteomics is inherently affected by missingness of the data in which case many of the proteins corresponding to transcripts seen in the mRNA dataset for example may be missing. How is this accounted for?

We agree that, as with any large scale datasets, conclusions cannot be drawn on missing data. In the correlative analyses, we have used data on only those genes/proteins that are common between the datasets used in the analysis, and thus contain the datapoints required to perform the correlations. In the materials and methods section, we have specified the selection criteria and numbers of genes/proteins included in each analysis.

None of the proteomics findings have been orthogonally validated except ACO2 and MDH2.

We have performed additional validations and present them as additional data in the revised manuscript.

While single standards are not feasible in mass spectrometry discovery analyses containing >40 000 peptides/sample, we used a global standard sample in each sample analysis queue to make sure that instrument and sample handling is repeatable and performing in correct level.

We have now performed a validation study with targeted mass spectrometry method for MicroLC-MSTrap using stable isotope labeled peptide standards for absolute quantification for 4 different proteins. The method and peptide information are provided in the materials and methods section, and in Supplementary Table 15. The results are presented in Supplementary figures 11 and 18. Discussion of these results has also been added to the manuscript text. This analysis confirmed our findings and provides a good basis for additional research based on the results of this study.

Here again it is unclear what is the role of increased ACO2 in PCa. It is well known that zinc transporters regulate ACO2 function and that Zn is an inhibitor of ACO2 activity. What does increase in ACO2 mean? What does progressive change in MDH2 mean? None of these have been addressed.

In this study, we identify alterations and pathways of interest from large scale data. We validated our discovery proteomics results by assessing the TCA cycle protein changes in more detail, and suggest that they may be functionally relevant for prostate cancer. The functional significance and the mechanisms of the detected and verified alterations need to be addressed in more detail with experimentation falling beyond the scope of this study.

In summary, there is no evidence presented to state that proteomics data in this case has uncovered driver events in prostate cancer progression as stated by the authors in the abstract.

In this work we have identified previously unknown alterations in prostate cancer both at protein expression, miRNA regulation, and pathway level, and provide a valuable proteomics resource for the community for further studies. However, we agree with the reviewer that the term driver event can be debated without further data on individual events, and thus we have removed the term from the abstract.

REVIEWERS' COMMENTS:

Reviewer #1 (Remarks to the Author):

In the revised manuscript the authors have addressed the issues raised in the initial review. The paper is acceptable.

Reviewer #2 (Remarks to the Author):

The authors have addressed most of the concerns of this reviewer. The authors are encouraged to include the description of CRPC samples they have used in the main text.

Latonen et al. - Response to reviewers' final comments

REVIEWERS' COMMENTS:

Reviewer #1 (Remarks to the Author):

In the revised manuscript the authors have addressed the issues raised in the initial review. The paper is acceptable.

Response: N/A

Reviewer #2 (Remarks to the Author):

The authors have addressed most of the concerns of this reviewer. The authors are encouraged to include the description of CRPC samples they have used in the main text.

Response: We have included a better description of the CRPC samples in the first paragraph of the results section.